# Novel Osteogenic and Easily Handled Endodontic Calcium Silicate Cement Using Pluronic F127 Hydrogel

**DOI:** 10.3390/ma15196919

**Published:** 2022-10-06

**Authors:** Jeong-Hyun Ryu, Jiyeon Roh, Utkarsh Mangal, Kwang-Mahn Kim, Sung-Hwan Choi, Jae-Sung Kwon

**Affiliations:** 1Department of Orthodontics, Institute of Craniofacial Deformity, Yonsei University College of Dentistry, Seoul 03772, Korea; 2Forensic Science Training and R&D Center, National Forensic Service, Wonju 26460, Korea; 3Department and Research Institute of Dental Biomaterials and Bioengineering, Yonsei University College of Dentistry, Seoul 03772, Korea; 4BK21 PLUS Project, Yonsei University College of Dentistry, Seoul 03772, Korea

**Keywords:** calcium silicate cement, pluronic F127 hydrogel, osteogenic property, endodontic filler

## Abstract

Calcium silicate cement (CSC) is widely used as an endodontic material in clinical applications such as direct pulp capping, pulpotomy, or root canal. CSC has good biocompatibility, sealing properties, and the ability to enhance hard tissue regeneration. However, the disadvantage of CSC is the difficulty in handling when placing it into endodontic tissue due to the long setting time. Several attempts have been made to improve handling of CSC; however, these methods were limited by osteogenic properties. To overcome such a disadvantage, this study investigated the use of Pluronic F127 (F127) for the development easy-to-handle novel endodontic CSCs with osteogenic properties. In this case, different concentrations of F127 (5%, 10%, 20%, 30%, and 40%) were implemented to generate CSC specimens H5, H10, H20, H30, and H40, respectively. Calcium ion was continuously released for 28 days. In addition, each group resulted in apatite formation for 28 days corresponding to calcium ion release. The concentration of F127 showed opposite relationships with water solubility and compressive strength. The H20 group showed a high level of osteogenic activity compared to other groups at 14 days. Mineralization of the H20 group was higher than that of the other groups. This study indicates that the novel F127-based hydrogel with CSC can potentially be used as endodontic filler.

## 1. Introduction

Initial development of dental filling materials includes materials such as amalgam, which was quickly replaced by polymethyl methacrylate, bisphenol-A glycidyl methacrylate, and triethylene glycol dimethacrylate that were introduced in 1960s, as the amalgam had limitation with use of the mercury. For enhancement of physical, chemical, and bioactivity, inorganic fillers were then added to the inorganic matrix of polymer, which resulted in development of dental composite materials that we commonly use till today [1,2,3].

Calcium silicate cement (CSC), which is well known as the main component of mineral trioxide aggregate, is used as an endodontic material [4]. CSC has been applied in a number of endodontic procedures such as pulp capping, partial pulpotomy, apexification, perforation repair, and root canal filling [5,6,7]. As revealed in these applications, CSC has sealing properties, good biocompatibility and bioactivity, and the ability to promote the differentiation of stem cells into odontoblasts or osteoblasts [8,9]. Commercial CSC is often mixed with liquid such as distilled water (DW), making it difficult to manipulate and handle after application because of the long setting time [10]. Therefore, many researchers have tried to overcome this disadvantage of CSC by using different carriers. In a previous study, polymer-based CSC was developed for a reduction in setting time and directly applied in the clinical environment [11]. However, the polymer-based CSC showed severe toxicity with poor physical properties and limited osteogenic properties [12,13].

A hydrogel is a network of polymer chains that are hydrophilic, usually with over 90% water content, and it possesses a degree of flexibility very similar to natural tissue [14,15]. Hydrogels are widely used as a scaffold in tissue engineering and regenerative medicine for mimicking the 3D microenvironment of cells [16]. Among the various hydrogels, environmentally sensitive hydrogels have the ability to sense changes in pH, temperature, or the concentration of a metabolite, releasing their load as a result of such a change [17,18].

Pluornic F127 (F127) is a triblock copolymer consisting of poly(ethylene oxide)/poly(propylene oxide)/poly(ethylene oxide) (PEO_100_–PPO_65_–PEO_100_), and it represents one of the most important thermoreversible hydrogels [19]. F127 has well known advantages of nontoxicity and water solubility, making it suitable for the development of hydrogels [20,21]. The F127 hydrogel undergoes a sol-to-gel transition with an increase in temperature [22]. Therefore, F127 gels have been widely investigated as a topical drug delivery carrier, for bone regeneration, and for wound healing [23,24,25].

In this study, we investigated the possible development of novel CSC endodontic material using F127 hydrogel. The combination with F127 hydrogel took advantage of the thermoreversible properties so as to instantly set the mixture according to body temperature. Moreover, F127 is well known as a biocompatible material in the biomedical field, enabling the osteogenic properties of CSCs to be maintained without showing toxicity.

To evaluate CSC products, we evaluated the calcium ion release, pH, water solubility, compressive strength, and bioactivity. In addition, we performed cytotoxicity tests, alkaline phosphatase staining and activity, and Alizarin Red S staining. The purpose of this study was to evaluate the physicochemical and biological properties of the novel combination of CSC with a thermoreversible hydrogel, F127. The null hypothesis is that hydrogel-based CSC would not significantly differ in terms of physical, chemical, and biological properties from the original CSC.

## 2. Materials and Methods

### 2.1. Materials

F127 powder, dimethyl sulfoxide (DMSO), L-ascorbic acid, β-glycerol phosphate, 3-[4,5-diemthyl-thiazol-2-yl]2,5-diphenyltetrazolium bromide, alizarin red s, cetylpyridinium chloride, ammonium hydroxide solution, and alkaline phosphatase (ALP) staining were all purchased to Sigma Aldrich (St. Louis, MO, USA). Simulated body fluid was purchased to Biosesang Co. (Seongnam-si, Korea). SensoLyte pNPP ALP assay kit was purchased to Anapspec (Fremont, CA, USA). Human mesenchymal stem cells (hMSCs) were purchased from Lonza (Basel, Switzerland). Dulbecco’s modified eagle medium (DMEM, High Glucose, GultaMAX^TM^), fetal bovine serum (FBS), and 1% antibiotics/antibiotic (AA), and Pierce^TM^ Bicinchoninic acid (BCA) Protein Assay kit were all purchased to Thermo Fisher Scientific (Waltham, MA, USA).

### 2.2. Preparation of F127 Hydrogel-Based CSC

The preparation of hydrogel referred to the cold method that was described in a previous study [26]. The samples were prepared by dissolving Pluronic F127 powder that were purchased from Sigma Aldrich (St. Louis, MO, USA) at various concentrations (5%, 10%, 20%, 30%, and 40% *w*/*v*) in distilled water at 4 °C overnight to facilitate dissolution of the polymer. The samples were stored 24 h and mixed with calcium silicate cement (Union, Seoul, Korea) in 1:0.3 ratios (codes: H5, H10, H20, H30, and H40). Calcium silicate cement mixed with distilled water was used as a control (MDW). 

### 2.3. Calcium Ion Release

A ring-shaped mold with 5 mm diameter and 2 mm height was prepared. After mixing each concentration of F127 and CSC, the samples were filled in the ring-shaped molds and set for 24 h at 37 °C. Next, the set samples were polished by removing excess material from the sides after removing them from the mold. The samples were immersed in 10 mL of distilled water adjusted to pH 7.4 and stored in a 37 °C water bath. Next, calcium ion levels were measured using a calcium ion electrode (Thermo Fisher Scientific) for 28 days.

### 2.4. Apatite Mineralization Assessment 

The apatite formation of the prepared hydrogels was evaluated using SBF, which was described in a previous study [27]. The samples of 10 mm diameter and 1 mm height were immersed in 15 mL of SBF at 37 °C for 28 days. After extraction from the SBF, samples were rinsed with double-distilled water and dried in an oven at 50 °C. The surface of the samples was observed using a scanning electron microscope and an energy-dispersive X-ray spectrometer (JEOL-7800F, JEOL, Tokyo, Japan). X-ray diffraction was also performed on these samples to identify any hydroxyapatite deposited on the samples after SBF tests.

### 2.5. Water Solubility 

The water solubility was determined in accordance with ISO 6876 [28]. Disc-shaped metal molds were prepared with 20 mm diameter and 1.5 mm height. After mixing each concentration of F127 and CSC, the samples were filled in the ring-shaped molds and set for 24 h at 37 °C. Subsequently, two specimens were weighed and then placed in a shallow glass dish, which was filled with 50 mL of distilled water and stored in a water bath (37 °C and 50% relative humidity). After being immersed in distilled water for 24 h, the samples were removed, and the surfaces of the two specimens were rinsed with 3 mL of fresh water on a shallow glass dish, which was then placed in a drying oven without boiling at 110 °C. The dried dishes were left to cool down at 25 °C, and then weighed. The solubility test was repeated three times.
$$WS\;\left(mg\right)=m_{0}-m_{1}$$

Here, WS is the water solubility, $m_{0}$ is the initial weight in fixation to 2 g, and $m_{1}$ is final weight after immersed in 24 h and dried.

### 2.6. Compressive Strength

The compressive strength was performed in accordance with ISO 9917-1 as there is no established standard method [29]. A metal mold of 4 mm diameter and 6 mm height was prepared. After mixing the samples, the materials were filled in the mold, before tightening a screw clamp over 120 s. These materials in the mold were placed in a water bath (37 °C and 50% relative humidity) for 24 h, before immersing them in distilled water for 24 h. Next, the compressive strength test was performed using a universal testing machine (Model 5942; Norwood, Instron, MA, USA) at a crosshead speed of 1.0 mm/min. The compressive load was recorded until a loading fracture point was reached. The maximum load at loading failure was used to calculate the compressive strength of the CSC samples using the equation described in ISO 9917-1.
$$\sigma =\frac{4P}{\pi d^{2}}$$

Here, *σ* is the compressive strength, *P* is the maximum applied load (N), and *d* is the mean diameter of the cylindrical samples (mm).

### 2.7. Cell Culture

hMSCs were cultured in DMEM supplemented with 10% FBS and 1% AA at 37 °C in 5% CO_2_. To make the osteogenic medium, 50 μg/mL l-ascorbic acid and 10 mM β-glycerol phosphate were supplemented to the culture medium. The hMSCs were cultured for 4–6 passages in this study.

### 2.8. Cell Viability

To confirm the cell viability of the control and experimental groups, we performed an MTT assay according to ISO 10993-5 [30]. Each experimental group consisted of a ratio of weight to volume of 0.2 g/mL, while the control group consisted of a ratio of area to volume of 3 cm^2^/mL, 15 mm diameter, and 2 mm height in accordance with ISO 10993-12 [31]. The control and experimental groups were immersed in cell culture medium, and then placed in a shaking incubator at 37 °C for 24 h. After 24 h, the supernatant was collected using a 0.2 μm filter. The hMSCs were seeded in 1 × 10^4^ cells/well in 96-well plates in 100 μL of culture medium. The medium was changed for extraction, and then the extracts were removed after 24 h. Next, 50 μL of MTT solution (1 mg/mL, 3-[4,5-dimethylthiazol-2-yl]-2,5-diphenyltetrazolium bromide, Sigma Aldrich) was added for 2 h, after which it was removed, leaving behind purple formazan. The purple formazan was dissolved in DMSO. The microplates were read using a microplate reader (Epoch, Biotek, VT, USA) at 570 nm. The tests were repeated three times, and all results were calculated as percentages with respect to the negative control.

### 2.9. In Vitro Osteogenic Properties

To determine the initial osteogenic expression of ALP, 1 × 10^5^ hMSCs were seeded in 12-well plates. After 24 h, the medium was changed to osteogenic medium for the extraction of control and experimental groups, which was replaced every 2–3 days. After culturing for 14 days, ALP staining (Sigma Aldrich) was performed according to the manufacturer’s instructions. In addition, ALP activity was determine using a SensoLyte pNPP alkaline phosphatase assay kit according to the manufacturer’s instructions. The total protein was normalized using a Pierce^TM^ BCA Protein Assay kit according to the manufacturer’s instruction.

To perform the mineralization after culturing for 28 days, Alizarin Red S staining (ARS) was performed in previous study [32]. The cells were fixed with 70% ice-cold alcohol for 1 h. The fixed cells were stained for 15 min with 2% ARS solution, pH 4.2 adjusted to ammonium hydroxide solution at 25 °C. After washing three times with distilled water, 10% cetylpyridinium chloride solution was added, and the absorbance of the solution was measured at 560 nm. The tests were performed three times.

### 2.10. Statistical Analysis 

Experimental data were processed using one-way analysis of variance followed by Tukey’s post hoc analysis (SPSS, Chicago, IL, USA); *p*-values under 0.05 were considered significant. Error bars represent the mean ± standard deviation.

## 3. Results

### 3.1. In Vitro Characterization of F127-Based Hydrogel with CSC

As shown in Figure 1, calcium ions in each group were released for 28 days in distilled water. The calcium release pattern of groups H5 to H40 was characterized by a burst on day 1, followed by continuous release until 7 days, and sustained release until 28 days. On the other hand, calcium ions in the MDW group were released in a burst released on day 1, followed by a decrease from 4 and 7 days, and then sustained release until 28 days. However, the calcium release pattern of the MDW group was shown to decrease in comparison with the F127 + CSC groups (*p* < 0.05).

Considering that calcium ions were released in all cases, the precipitation of apatite formation could be observed through exchange with various ions from the SBF to determine bioactivity, as shown in Figure 2A. The formation of crystalline aggregates was revealed on the surface of each group through the exchange of calcium ions from the CSC and phosphate ions from the SBF. Furthermore, the surface of each group was analyzed using X-ray diffraction, which revealed the phase of hydroxyapatite corresponding with JCPDS file 09-432, as shown in Figure 2B.

### 3.2. Physical and Mechanical Properties of F127-Based Hydrogel with CSC

The water solubility of each group is shown in Figure 3A. There was no significant difference in water solubility between the MDW group and H5 group; however, it increased with the concentration of F127 from H10 to H40. The compressive strength of each group is shown in Figure 3B. The compressive strength of the H5 and H10 groups were not significantly different in comparison with the MDW group. However, the compressive strength of the H30 and H40 groups were the lowest in comparison with the other groups (*p* < 0.05). Taken overall, the concentration of F127 showed opposite relationships with water solubility and compressive strength.

### 3.3. Cell Viability

In terms of biological performance, the cell viability in all cases was approximately 10%, with no significant difference among groups. Therefore, we diluted the samples twofold, which led to a cell viability over 80% for all groups (Figure 4).

### 3.4. Osteogenic Properties

In accordance with the cell viability and considering the clinical environment, we chose the H20 and H30 groups for evaluating osteogenic differentiation through ALP staining (Figure 5A). The ALP activity of the H20 group was significantly higher in comparison with the other groups (*p* < 0.05), revealing the darkest purple stain. Subsequently, calcium deposition was evaluated through ARS staining (Figure 5B). The calcium deposition (*p* < 0.05) and the mineralization of the H20 group were higher than those of the MDW and H30 groups.

## 4. Discussion

CSC is a well-known endodontic material used in direct pulp capping, partial pulpotomy, or root filling. In addition, CSC can be effectively used to regenerate hard tissue due to its bioactive and osteogenic properties. CSC is a biocompatible material with sealing properties; however, it suffers from a long setting time and difficult handling, restricting its use in the clinical environment. Therefore, many researchers have developed polymer and CSC composite materials so as to attain a shorter setting time. The most representative polymer and CSC composite materials implemented a light curable-based resin; however, this affected the cement negatively by increasing the risk of shrinkage and reducing mechanical strength [33]. From a biological perspective, light-curable resin also led to increased cytotoxicity [34]. Accordingly, the aim of this study was to develop a biocompatible polymer of the hydrogel for combination with CSC. Hydrogels are hydrophilic polymers that generate a three-dimensional polymeric network upon storing a large amount of water. Their water-rich structure is similar to biological tissue as a biomimetic application, and they are commonly used in tissue regeneration or as drug delivery systems. Hydrogels created using various biocompatible polymers have demonstrated hard tissue regeneration with bioceramics [35]. In addition, F127 approved by the FDA also exhibits thermosensitivity, forming a gel at body temperature [36]. Considering this advantage of F127-based hydrogel, it was combined with CSC in this study, followed by an evaluation of its mechanical and biological properties linked to potential hard tissue regeneration.

The calcium ion release of each experimental group was analyzed over 28 days. The pH of the experimental group was maintained as alkaline throughout this period. F127-based hydrogel plays a role in the controlled release of drugs or therapeutic ions. Borges et al. and Hou et al. performed in vitro studies, revealing that F127-based hydrogel loaded with bioceramics or drug was continuously released [35,37]. To evaluate the bioactivity of the composite hydrogel and CSC, the deposition of hydroxyapatite on the surface of each group was recorded following SBF immersion. The F127-based hydrogel with CSC could continuously release calcium ions over 28 days, enabling their exchange with the phosphate ions in SBF [38,39].

In terms of physical and mechanical properties, the F127-based hydrogel with CSC showed an increase in water solubility with the concentration of F127, as a function of its hydrophilicity [40]. In contrast, the compressive strength decreased with the concentration of F127. Although as the endodontic material, the material will not undergo clinical situation where large amount of compressive strength would be applied on the material, the compressive strength is still important feature as the low compressive strength would results in the hydrogel structure becoming unstable during drug release due to its poor mechanical strength in physiological conditions [41].

All groups were cytotoxic at 100% concentration; however, a twofold dilution led to a cell viability in excess of 80%. At 50% concentration of extract, H20 showed decrease in average cell viability percentage compared to H10, though there was no statistical significance (*p* > 0.05). However, there was a statical significant increase in cell viability for H30 compared to H20 (*p* < 0.05) which may be due to the more sustained release of ions from CSC along with proliferative/differentiating effects on cells as indicated by latter experiments results. H Nabavizadeh et al. reported that a threefold dilution of CSC exposed to fibroblast for 3 days exhibited cytotoxicity due to the high calcium ion concentration in an alkaline environment [42].

On the basis of the cell viability, we chose the H20 and H30 groups for further characterization. These samples with a high concentration of F127 were characterized by high water solubility and low compressive strength, which has been shown to sustain the release of drug or therapeutic ions [43]. However, Ye et al. reported that hydrogels led to an increase in the micellar number density and a decrease in the substitutional space of aqueous pores within the micelles with increasing polymer content [44], due to poor mechanical strength.

In terms of osteogenic properties, the ALP expression of the H20 group was significantly higher than that of the other groups (*p* < 0.05). Furthermore, the effect of calcium ion release from the H20 group also led to effective mineralization (*p* < 0.05). CSC was previously shown to significantly promote osteogenesis both in vivo and in vitro through the release of calcium ions [45]. The osteogenic differentiation of osteoblast-like cells is a key stage of bone regeneration. ALP expression is an early indicator of bone formation [46], shifting osteoblasts to a differentiation state [47,48] and stimulating mineralization in osteogenesis through an increase in inorganic phosphate [49].

With respect to the limitation of this study and for the future possible work, the study was limited to in vitro results related to possible hard tissue regeneration and future experiments on other in vitro studies such as physical, chemical, thermal, and mechanical properties of the materials, in vivo or clinical results relate to the application of F127 composite with CSC may be warranted. Still, the study indicated the potential of the application of F127-based hydrogel with CSC for the enhancement of hard tissue regeneration. 

## 5. Conclusions

We successfully developed a novel CSC-based endodontic material combined with F127-based thermoreversible hydrogel. The controlled release of calcium from CSCs in F127-based hydrogel was observed over 28 days, resulting in hydroxyapatite formation. However, a high concentration of F127 resulted in increased water solubility and lower compressive strength. Accordingly, the H20 group was indicated as the best candidate for enhancing osteogenic properties, as demonstrated by the ALP expression and calcium deposition of hMSCs. Within the limitations of this in vitro study, necessitating further preclinical and clinical studies, it can be concluded that F127-based hydrogel with CSC may be feasible as an effective endodontic material.

## Figures and Tables

**Figure 1 materials-15-06919-f001:**
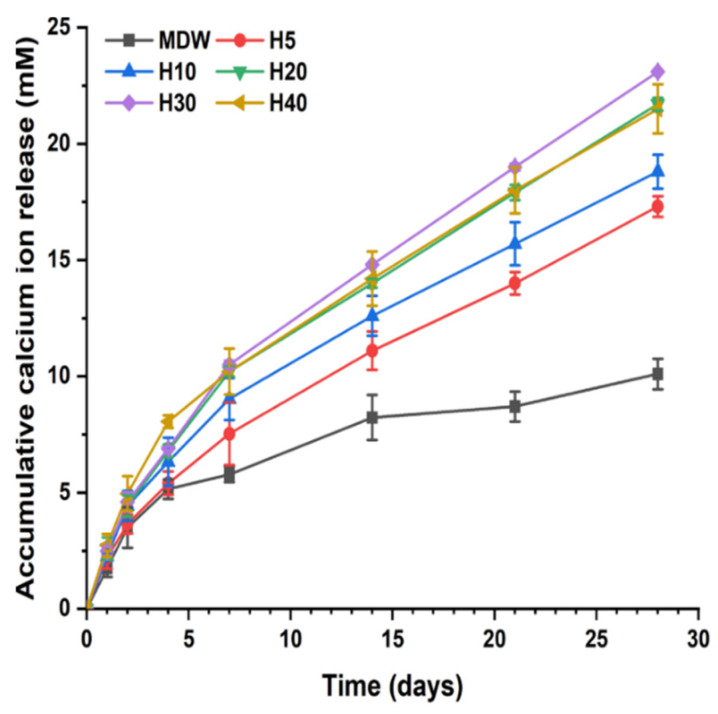
Calcium ion release of each experimental group in distilled water over 28 days. The calcium ions of the experimental group were proportionately released from days 7 to 28 in contrast to the MDW group.

**Figure 2 materials-15-06919-f002:**
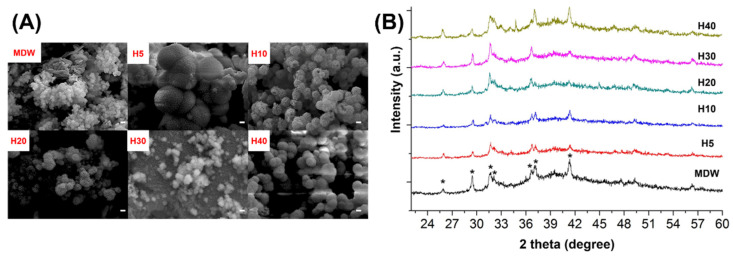
(**A**) Representative images of apatite formation in control and experimental groups after immersion in SBF for 28 days (scale bar = 1 μm). (**B**) X-ray diffractograms highlighting hydroxyapatite phase.

**Figure 3 materials-15-06919-f003:**
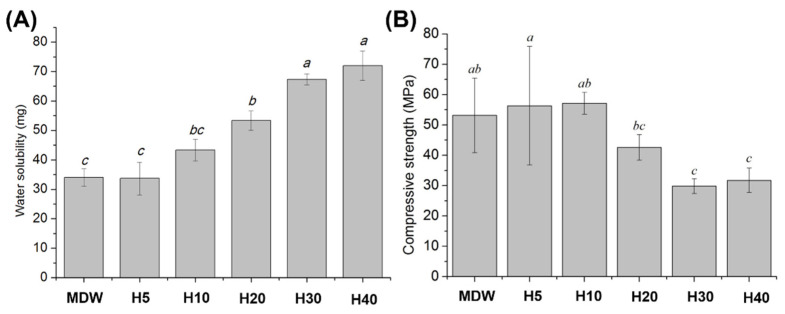
The physical and mechanical properties of control and experimental groups: (**A**) water solubility; (**B**) compressive strength. The F127 concentration showed opposite relationships with water solubility and compressive strength. Different small case letters indicate statistically significant differences between the groups with analysis of variance and Tukey’s post-hoc.

**Figure 4 materials-15-06919-f004:**
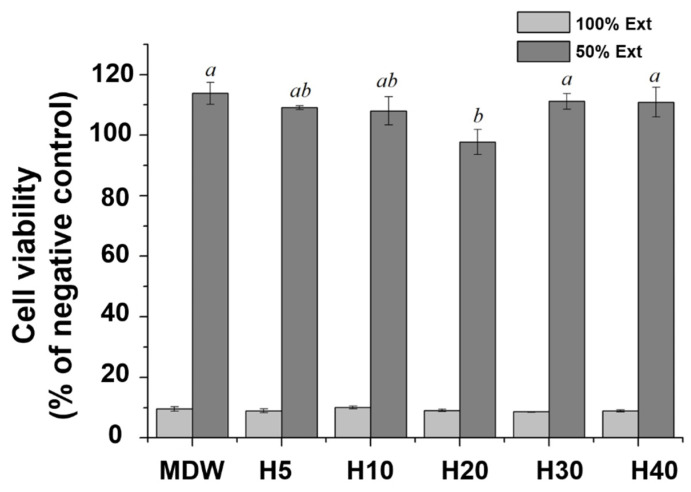
Cell viability of control and experimental groups. The 100% concentration was cytotoxic in all cases, whereas the 50% concentration featured cell viability exceeding 80%. Different small case letters indicate statistically significant differences between the groups with analysis of variance and Tukey’s post-hoc.

**Figure 5 materials-15-06919-f005:**
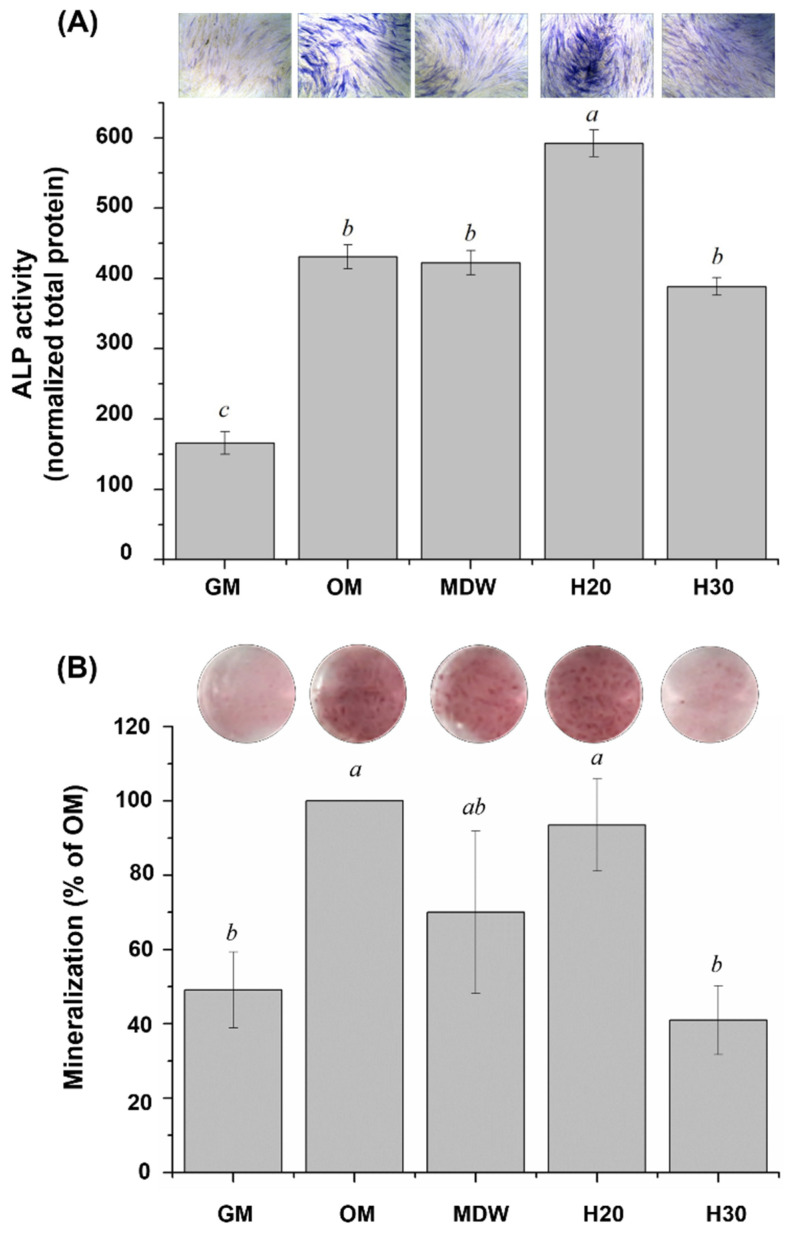
Osteogenic properties of control and experimental groups. (**A**) ALP staining and activity, highlighting significant difference of H20 group (*p* < 0.05). (**B**) Mineralization, highlighting a higher but insignificant difference of H20 group. Different small case letters indicate statistically significant differences between the groups with analysis of variance and Tukey’s post-hoc. GM; growth medium, OM; osteogenic medium.

## Data Availability

The data presented in this study are available on request from the corresponding authors.

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
