# Peer review of "Novel Osteogenic and Easily Handled Endodontic Calcium Silicate Cement Using Pluronic F127 Hydrogel"

_materials, 2022, doi:10.3390/ma15196919_

Round 1
Reviewer 1 Report
A. Introduction part (first paragraph) is not significant. Please add some dental history and add recommended articles which are given below.
1. Dental restorative composite materials: A review. J Oral Biosci 2019; 61(2): 78-83. https://doi.org/10.1016/j.job.2019.04.001
2. Investigation of the physical, mechanical and thermal properties of nano and microsized particulate-filled dental composite material. J Compos Mater 2020; 54(19): 2623-2633. https://doi.org/10.1177/0021998320902212
3. Analytic hierarchy process‐technique for order preference by similarity to ideal solution: A multi criteria decision‐making technique to select the best dental restorative composite materials. Polym Compos 2021; 42 (12): 6867-6877. https://doi.org/10.1002/pc.26346
B. Add water solubility formula
C. Also add future scope of the study.
D. What about of physical, chemical, thermal, mechanical properties of the current study?
E. The compressive strength results is not meaningful. Why?
F. Figure 4; Cell viability is decreased till H20 and afterwards it increased. why
G. Add citations in section 2.1 to 2.7
H. what is tempearture range in whole experiment?
Author Response
|
Reviewer 1 |
|
|
Q1 |
Introduction part (first paragraph) is not significant. Please add some dental history and add recommended articles which are given below. |
|
A1 |
Thank you for your comments with references. We now have considered the references and added below note in Introduction (red in colour); “Initial development of dental filling materials include materials such as amalgam, which was quickly replaced by polymethyl methacrylate, bisphenol-A glycidyl methac-rylate, and triethylene glycol dimethacrylate that were introduced in 1960s, as the amalgam had limitation with use of the mercury. For enhancement of physical, chemical, and bioactivity, inorganic fillers were then added to the inorganic matrix of polymer, which resulted in development of dental composite materials that we commonly use till today [1-3].” |
|
Q2 |
Add water solubility formula |
|
A2 |
Thank you for your valuable comment. We now have added the formula of water solubility in Materials and Methods, 2.5 Water Solubility (red colour) as below; |
|
Q3 |
Also add future scope of the study. |
|
A3 |
Thank you for your comments. We now have included possible future work related to this study (and related to limitation of this study) in end of Discussion, as below (red colour) With respect to the limitation of this study and for the future possible work, the study was limited to in vitro results related to possible hard tissue regeneration and future experiments on other in vitro studies such as physical, chemical, thermal, and mechanical properties of the materials, in vivo or clinical results relate to the application of F127 composite with CSC may be warranted. Still, the study indicated the potential of the application of F127-based hydrogel with CSC for the enhancement of hard tissue regeneration. |
|
Q4 |
What about of physical, chemical, thermal, mechanical properties of the current study? |
|
A4 |
Thank you for your valuable comments. In this study, our aim was to investigate possibilities of incorporating F127 matrix with CSC and consider effect of hard tissue regeneration at in vitro level. Therefore, other studies that may related to properties of endodontic materials but not directly related to hard tissue regenerations are not considered and may be warranted as future studies. Limitation with these and possible future studies are indicated as above comment, as below; With respect to the limitation of this study and for the future possible work, the study was limited to in vitro results related to possible hard tissue regeneration and future experiments on other in vitro studies such as physical, chemical, thermal, and mechanical properties of the materials, in vivo or clinical results relate to the application of F127 composite with CSC may be warranted. Still, the study indicated the potential of the application of F127-based hydrogel with CSC for the enhancement of hard tissue regeneration. |
|
Q5 |
The compressive strength results is not meaningful. Why? |
|
A5 |
Sorry for the confusion in writing the manuscript. Although as the endodontic material, compressive strength may not to be too high, it is still important features of the material as it can determine hard tissue regeneration properties as CSC is contained within the hydrogel. Therefore, we have revised Discussion (in red colour) to indicate this as below; Although as the endodontic material, the material will not undergo clinical situation where large amount of compressive strength would be applied on the material, the compressive strength is still important feature as the low compressive strength would results in the hydrogel structure becoming unstable during drug release due to its poor mechanical strength in physiological conditions [41]. |
|
Q6 |
Figure 4; Cell viability is decreased till H20 and afterwards it increased. Why |
|
A6 |
Thank you for your valuable comment. We expected that such result is probably due to sustained release of ions along with cell differentiation/proliferation effects as below (red colour in Discussions); At 50% concentration of extract, H20 showed decrease in average cell viability percentage compared to H10, though there was no statistical significance (p > 0.05). However, there was a statical significant increase in cell viability for H30 compared to H20 (p < 0.05) which may due to the more sustained release of ions from CSC along with prolifera-tive/differentiating effects on cells as indicated by latter experiments results. |
|
Q7 |
Add citations in section 2.1 to 2.7 |
|
A7 |
Thank you for your valuable comment. We have now added the citation throughout the text, especially in section 2.2 to 2.8. |
|
Q8 |
what is temperature range in whole experiment? |
|
A8 |
Thank you for your valuable comments. We performed the whole experiment in 25 °C temperature exception for biological properties. We now have included detail in in Materials and Method. |
Reviewer 2 Report
Remarks:
Kindly add appropriate citations for the methodological details at all places .
Abbreviations need to be rechecked and shall be provided as expanded terms at their first use in the manuscript.
Author Response
|
Reviewer 2 |
|
|
Q1 |
Kindly add appropriate citations for the methodological details at all places. |
|
A1 |
Thank you for your comments. We have now added the citation throughout the text, especially in section 2.2 to 2.8 (red colour). |
|
Q2 |
Abbreviations need to be rechecked and shall be provided as expanded terms at their first use in the manuscript. |
|
A2 |
Thank you for your valuable comments. We have now confirmed the abbreviations or expanded terms. |
Reviewer 3 Report
I find the manuscript very interesting and necessary. The authors' research has been carefully described and presented, but I have two comments
1. in Chapter 2, Materials and methods, I did not find information about the tested material, but only information about research methods
2. The manuscript has been submitted as an "article" and in my opinion the Authors should follow the order established by the editorial office for this type of work. The manuscript has virtually no introduction. In this part, the authors provide information on the materials tested and the tests performed. Please expand the general part a bit, and include the one about materials in the second chapter
Author Response
|
Reviewer 3 |
|
|
Q1 |
I find the manuscript very interesting and necessary. The authors' research has been carefully described and presented, but I have two comments |
|
A1 |
Thank you for review of our manuscript. We have replied to your comments as below. |
|
Q2 |
In Chapter 2, Materials, and methods, I did not find information about the tested material, but only information about research methods |
|
A2 |
Thank you for your comments and sorry for confusion. We have listed tested materials in 2.1 Materials section, and preparations from such materials to form F127 hydrogel-based CSC have been described in 2.2 Preparation of F127 Hydrogel-Based CSC. To avoid confusions, we have now reorganized text (in red colour). Hope this would clarify your question. |
|
Q3 |
The manuscript has been submitted as an "article" and in my opinion the Authors should follow the order established by the editorial office for this type of work. The manuscript has virtually no introduction. In this part, the authors provide information on the materials tested and the tests performed. Please expand the general part a bit, and include the one about materials in the second chapter |
|
A3 |
Thank you for your comments. We have now modified Introduction to include more general information about this material. Also, 2.1 Materials section has been modified and reorganized along with other parts of Materials and Methods, in accordance to format provided by MDPI Materials template. Thank youyou’re your input. |